# Isolation and Characterisation of Bacteriophages with Activity against Invasive Non-Typhoidal *Salmonella* Causing Bloodstream Infection in Malawi

**DOI:** 10.3390/v13030478

**Published:** 2021-03-15

**Authors:** Ella V. Rodwell, Nicolas Wenner, Caisey V. Pulford, Yueyi Cai, Arthur Bowers-Barnard, Alison Beckett, Jonathan Rigby, David M. Picton, Tim R. Blower, Nicholas A. Feasey, Jay C. D. Hinton, Blanca M. Perez-Sepulveda

**Affiliations:** 1Clinical Infection, Microbiology and Immunology, Institute of Infection, Veterinary and Ecological Sciences, University of Liverpool, Liverpool L69 7ZB, UK; ella.rodwell@phe.gov.uk (E.V.R.); mwenner@liverpool.ac.uk (N.W.); caisey.v.pulford@phe.gov.uk (C.V.P.); yueyi.cai@liverpool.ac.uk (Y.C.); arthurbowbar@hotmail.com (A.B.-B.); jay.hinton@liverpool.ac.uk (J.C.D.H.); 2Biomedical Electron Microscopy Facility, University of Liverpool, Liverpool L69 7ZB, UK; alib@liverpool.ac.uk; 3Department of Clinical Sciences, Liverpool School of Tropical Medicine, Liverpool L3 5QA, UK; Jonathan.Rigby@lstmed.ac.uk (J.R.); Nicholas.Feasey@lstmed.ac.uk (N.A.F.); 4Malawi Liverpool Wellcome Trust Clinical Research Programme, The College of Medicine, University of Malawi, Blantyre 3, Malawi; 5Department of Biosciences, Durham University, Durham DH1 3LE, UK; david.m.picton@durham.ac.uk (D.M.P.); timothy.blower@durham.ac.uk (T.R.B.)

**Keywords:** Enteritidis, Typhimurium, Malawi, environmental phage

## Abstract

In recent years, novel lineages of invasive non-typhoidal *Salmonella* (iNTS) serovars Typhimurium and Enteritidis have been identified in patients with bloodstream infection in Sub-Saharan Africa. Here, we isolated and characterised 32 phages capable of infecting *S*. Typhimurium and *S*. Enteritidis, from water sources in Malawi and the UK. The phages were classified in three major phylogenetic clusters that were geographically distributed. In terms of host range, Cluster 1 phages were able to infect all bacterial hosts tested, whereas Clusters 2 and 3 had a more restricted profile. Cluster 3 contained two sub-clusters, and 3.b contained the most novel isolates. This study represents the first exploration of the potential for phages to target the lineages of *Salmonella* that are responsible for bloodstream infections in Sub-Saharan Africa.

## 1. Introduction

Worldwide, non-typhoidal *Salmonella* (NTS) *enterica* subspecies *enterica* serovars Typhimurium and Enteritidis (*S*. Typhimurium and *S*. Enteritidis) cause self-limiting enterocolitis in humans. However, in recent years novel lineages of NTS serovars Typhimurium and Enteritidis have been associated with increasing levels of invasive non-typhoidal *Salmonella* (iNTS) disease, mostly in Sub-Saharan Africa [1]. These bloodstream infections occur in immunocompromised individuals, resulting in about 50,000 deaths each year in Africa [1]. These lineages are characterised by genomic degradation, a distinct prophage repertoire, and multidrug resistance [2,3]. *S.* Typhimurium sequence type (ST) 313 has been widely studied [4,5] due to its association with iNTS disease, and D23580 is the representative strain for Lineage 2. Although *S.* Enteritidis is the other main serovar responsible for iNTS in Africa, this serovar has been primarily studied in relation to the implications in food industry. Several distinct and geographically linked lineages have been identified that cause iNTS (the West African, and Central and Eastern African lineages) or the global enterocolitis epidemic [3].

Phage-resistance mechanisms and other frequently carried genes involved in bacterial metabolism and pathogenesis increase bacterial fitness and support adaptation to different environments [6,7]. Prophages found in iNTS have been described in *S*. Typhimurium [8,9], and partially described in *S*. Enteritidis [3], for example *Salmonella* prophages Gifsy-1, Gifsy-2, BTP1, and Fels2-BT. Prophages play an important role in *Salmonella* virulence, increasing fitness and preventing infection by other phages [6,10,11]. Indeed, a novel phage exclusion system was identified encoded in *S.* Typhimurium D23580 prophage BTP1 [10], supporting the host to survive phage predation in the environment and drive niche specialisation. There are diverse anti-phage systems widely spread in bacteria [12,13], stressing the need to find phages that are capable of escaping these anti-phage mechanisms that protect African iNTS.

Moreover, the increase in antimicrobial resistant strains is driving the search for alternatives to widely used antibiotics. Here, we aimed to isolate and characterise phages capable of lysing the novel lineages of *S*. Typhimurium and *S*. Enteritidis associated with bloodstream infection. We sought phages on two continents, in both the UK and Malawi, with the objective of isolating a pool of phages with a different host range.

## 2. Materials and Methods

### 2.1. Cloning Procedures

All the bacterial strains, plasmids, and bacteriophages used in this study are described in Table 1, and oligonucleotides (primers) are described in Appendix A. DNA manipulations were carried out according to standard protocols [14] and to enzyme/kit manufacturer’s recommendations. Restriction enzymes and T4 DNA ligase were purchased from Thermo Fisher Scientific or New England Biolabs (NEB) and DNA purification and plasmid isolation kits were obtained from Bioline (Cat#BIO-52060 and Cat#BIO-52057, respectively).

For the construction of pEMG-based suicide plasmids [26], the DNA inserts of interest were digested with EcoRI and BamHI and ligated into pEMG, using the corresponding sites.

For the construction of pNAW1, two DNA fragments (~750 bp each) flanking the Gifsy-2 attachment site (*attB*^Gifsy-2 4/74^) of *S.* Typhimurium strain 4/74 were PCR amplified with primer pairs NW_1/NW_2 and NW_3/NW_4. The resulting fragments were fused together by overlap extension PCR [30], as primers NW_2 and NW_3 share a complementary stretch of 25 nucleotides. Finally, the resulting fragment (1529 bp) was digested and ligated into pEMG. Similarly, a suicide plasmid was constructed to knock out the *galE* gene (pNAW34): the flanking regions of *galE* were amplified with primer pairs NW_82/NW_83 and NW_84/NW_85 and the resulting fragments (~450 bp each) were fused by overlap extension PCR, digested and ligated into pEMG.

*S.* Typhimurium strain 4/74 carries the SopEΦ prophage that is flanked by a P4-like remnant prophage (hereafter called P4-like) [31]. Those prophage elements are absent in strain 14028s that was used to amplify the corresponding empty *attB* site with primers NW_65 and NW_65. After digestion, the resulting fragment (1531 bp) was ligated into pEMG. All the pEMG-derived plasmids were checked by Sanger sequencing using primers M13_-40_long and M13_rev_long and the Lightrun sequencing service (Eurofins Genomics).

### 2.2. Construction of Mutant Hosts

The anti-phage *brex* operon [32] (genes *STMMW_44361-STMMW_44431,* GenBank accession number FN424405.1) was deleted in the prophage-free strain *S.* Typhimurium D23580 ΔΦ (JH3949) by λ *red* recombination [27]. Electro-competent D23580 ΔΦ carrying the temperature inducible λ *red* plasmid pSIM5-*tet* [28] were prepared, as previously described [33]. Competent bacteria were electroporated with a PCR fragment obtained by PCR with primers Brex_del_Fw and Brex_del_Rv and pKD4 as template. After selection on kanamycin (Km) medium, recombinant D23580 ΔΦ ∆*brex*::Km mutants were obtained. The mutation was transferred into D23580 ΔΦ by transduction using the P22 HT *105/1 int-201* transducing phage [8,16]. Finally, the Km ^R^ cassette was removed with the flippase expressing plasmid pCP20-Gm [29], resulting in strain D23580 ΔΦ ∆*brex*::*frt* (JH4314). Finally, the mutation was confirmed by Sanger sequencing using the primers Brex_up and Brex_down.

For the scarless excision of prophages and for the deletion of *galE*, the technique based on the pEMG suicide was used [26]. Suicide plasmids carrying the appropriate empty *attB* or the ∆*galE* fragment were mobilised from *Escherichia coli* S17-1 λ*pir* into *S.* Typhimurium by conjugation and the resulting Km resistant meroploids were resolved with the pSW-2 plasmid, as described previously [8,27,34]. After successive prophage deletions, the prophage-free strain *S.* Typhimurium 4/74 ΔΦ (JH4180) was obtained: this strain lacked the Gifsy-1, Gifsy-2, ST64B, SopEΦ and P4-like prophage elements and was sequenced using the Illumina whole-genome sequencing service at MicrobesNG (Birmingham, UK).

### 2.3. Phage Isolation

Water samples were enriched with either *S.* Typhimurium or *S.* Enteritidis hosts (Table 2) for initial bacteriophage isolation. Further propagation and storage of isolated phages was carried out by using a susceptible *Salmonella* host (*S.* Typhimurium D23580 ∆Φ ∆*brex*) constructed from the iNTS strain D23580 [4] with all prophages removed [8], lacking in addition the BREX bacteriophage exclusion defence system [32].

Several locations were selected in the UK and Malawi for field sampling. The sampling locations in Blantyre, Malawi were around Mudi river [35], Naperi river, drains near food markets and Queen Elizabeth Central Hospital. The sampling locations in the UK were pond and river water near parks (Sefton Park, Liverpool, and Botanic Gardens, Southport) and the outflow from sewage treatment plants (River Alt, Liverpool, and River Wear, Durham).

Briefly, we collected 50 mL water samples that were first centrifuged at 7000 rpm (Eppendorf Centrifuge 5430R) at room temperature for 5 min. The enrichments were prepared by filtering 10 mL supernatant through 0.22 µm pore-size membranes (Star Lab Cat#E4780-1226) into 1:1 v/v LB 2X (2% tryptone, 1% yeast extract, 1% NaCl; pH 7.0) inoculated with the *Salmonella* host (1:20 *v*/*v*) grown overnight, and incubating with agitation (200 rpm) for 3 days at 37 °C. Then, 5 mL aliquots from each enrichment were centrifuged at 7000 rpm (Eppendorf Centrifuge 5430R) at room temperature for 15 min, and the supernatant filtered through 0.22 µm pore-size membranes. A 10-fold serial dilution was prepared using sterile LB for double-layer plaque assay [10,36] on either 90 mm or 12 × 12 cm Petri dishes, incubated overnight at 37 °C. Individual plaques with different morphologies were selected and purified repeating the serial dilution and plaque assay process twice.

Phage titration and storage were carried out using the susceptible *Salmonella* Typhimurium strain D23580 ∆Φ ∆*brex.* For storage, plaques were selected from plates and the phages resuspended in sterile LB with 1% chloroform, followed by vigorous shaking and centrifugation at 14,000 rpm (Eppendorf Centrifuge 5424) for 5 min at room temperature, and stored at 4 °C in the dark.

### 2.4. Strain Selection and Phage Host Range

The *S.* Typhimurium D23580 strain was used for phage isolation because it is a representative of ST313 lineage 2, the *Salmonella* variant responsible for iNTS epidemics in Malawi. Other relevant *Salmonella* strains were also included to expand the likelihood of phage isolation. The *S.* Enteritidis strains selected were representatives of four of the major clades, namely the Global Epidemic, Central/Eastern, and West African clades [3].

To determine the host range of the isolated bacteriophages, a spot test [37] was performed using phage stocks that contained between 10^10^ and 10^11^ pfu/mL. Briefly, bacterial lawns of different *Salmonella* hosts (Table 1) were prepared by diluting overnight bacterial cultures 40 folds with top-agar (0.5% agar in LB, 50 °C), and pouring over 1.5% LB-agar plates (4 mL top agar for 90 mm diameter or 8 mL for 12 × 12 cm Petri dishes). The plates were incubated for 24 h at 37 °C, and lysis formation recorded (Appendix A). This assay was also performed using characterised published (reference) phages (Table 1) and using between two and six biological replicates.

### 2.5. RAPD PCR

Randomly amplified polymorphic DNA polymerase chain reaction (RAPD PCR) was performed using the phage lysate as template. The phage lysate was prepared with 85 µL phage stock (10^10^–10^11^ pfu/mL), treated with 10 µL 10XDNase buffer (100 mM Tris-HCl, 25 mM MgCl_2_, pH 7.5) and 5 µL 10 mg/mL DNase I (SIGMA DN25), incubated at 37 °C shaking (300 rpm) for 60 min, followed by a second incubation at 75 °C for 10 min. Four oligonucleotides were designed [38] for PCR using MyTaq Red Mix 1X (Bioline, Cat#BIO-25043), according to the manufacturer’s recommendations. The oligonucleotides used were OPL5, RAPD5, P1 and P2 (Appendix A), as previously described [38]. The PCR products were analysed by 0.8% agarose-gel electrophoresis with 1 kb ladder (Bioline, Cat#H1-819101A), run at 100 V for 60 min, and visualised with Midori Green DNA/RNA staining (Nippon Genetics, Cat#MG06).

### 2.6. Phage DNA Extraction and Sequencing

Phage DNA was extracted from 410 µL of phage stock (5 × 10^10^–10^11^ pfu/mL) with 50 µL 10X DNase buffer (100 mM Tris-HCl, 25 mM MgCl_2_, pH 7.5), 20 µL 10 mg/mL DNase I (SIGMA DN25), and 20 µL of 1 mg/mL RNase A (SIGMA), incubated at 37 °C shaking (300 rpm) for 60 min, followed by a second incubation at 75 °C for 10 min. The lysate was treated with 10 µL 20 mg/mL Proteinase K (BIOLINE, Cat#BIO-37037) and purified using the Norgen Phage DNA Isolation Kit (cat. 46850) following manufacturer’s instructions. The purified DNA was eluted in 75 µL molecular grade water. DNA integrity was assessed by 1% agarose-gel electrophoresis, and DNA concentration was determined fluorometrically with dsDNA HS Assay kit (Qubit, ThermoFisher Scientific).

Phage DNA sequencing was carried out at Nu-Omics DNA sequencing research facility (Northumbria University, UK) using MiSeq (Illumina). Generated reads were quality checked using FastQC v0.11.5 (https://www.bioinformatics.babraham.ac.uk/projects/fastqc/, accessed on 9 May 2019), and quality trimmed using SEQTK v1.3-r106 (https://github.com/lh3/seqtk, accessed on 9 May 2019). Phage genomes were assembled using Unicycler v0.3.0b [39] with default parameters, and the quality of the assemblies were assessed with QUAST v4.6.3 [40].

Genomes that had more than one contiguous sequence were visualised in Artemis v10.2 [41], and circularised contigs were extracted for viral species analysis using BLAST [42]. Phages BPS7 and ER24 had multiple linear contigs, therefore BLAST was used to identify the viral contig for extraction and analysis. Phage assemblies were annotated using Prokka v1.12 [43] with--proteins/blastdb/Viral_Genomes/all_viral.faa, visualised using SnapGene v5.2 (Insightful Science, snapgene.com) and manually curated (Appendix A). PhageAI [44] is a tool used to determine lifecycle of phages with high accuracy. Phage genome assemblies were analysed using PhageAI v0.1.0 LifeCycleClassifier using default parameters.

All genomes were uploaded to GenBank with phage names assigned based on Adriaenssens and Brister (2017) [45]; however, common names were used for simplicity (Table 2).

### 2.7. Comparative Genomics and Phylogenetic Analyses

Gene sequence alignment of terminase large subunit (*terL*) was performed by ClustalO v1.2.4 [46] using default parameters, and a maximum likelihood tree was constructed using RAxML v8.2.8 [47], visualised in the interactive Tree of Life (iTOL) v4.2 [48] rooted at midpoint. Genomic similarities were calculated by pairwise nucleotide comparison with the Virus Intergenomic Distance Calculator (VIRIDIC) [49] with default parameters.

Genomes of phages belonging to the same cluster were analysed. A gene alignment was performed using Roary v3.11.0 [50] with default parameters for each cluster. A maximum likelihood phylogenetic tree was built using RAxML-NG v0.4.1 beta [47] with 100 bootstrap replicates to assess support, and visualised in iTOL v4.2 unrooted. Genome comparisons were performed using nucleotide alignment with MEGA v6.06 [51] and BLASTn [42], and visualised with Brig v0.95 [52].

### 2.8. Transmission Electron Microscopy (TEM)

Morphology of selected phages was determined by TEM [8], using 10 µL of phage stock (10^10^–10^11^ pfu/mL) pipetted onto carbon-coated copper 200-mesh grid (TAAB F077/050), and incubated at room temperature for 10 min. The grids were washed twice in distilled water for 2 min. To negative-stain the phages, 10 µL of 2% uranyl-acetate was added and incubated for 1 min. Grids were allowed to dry and TEM was performed on EI 120 kV Tecnai G2 Spirit BioTWIN transmission electron microscope.

## 3. Results and Discussion

### 3.1. Phage Isolation

To isolate a diverse pool of phages, we used water samples collected in the UK and Malawi, close to areas where *Salmonella* or other related bacteria would be expected to coexist, such as sewage plants or outlets. We enriched for phages that plated on *S.* Typhimurium or *S.* Enteritidis hosts (Table 2), and we selected plaques with different morphologies. The RAPD PCR assay was used to differentiate the phages, and 32 distinct phages were selected for whole genome sequencing (Appendix A). The 32 bacteriophages included 20 phages isolated from Malawi and 12 phages from the UK.

The majority of the phages (25 phages, 78%) were isolated using the phage-susceptible *S.* Typhimurium D23580 ΔΦ Δ*brex* mutant as a host. This mutant was constructed by removing all known ‘anti-phage’ encoding genetic features, which include all five prophages [8] and BREX bacteriophage exclusion defence system [32], from a well-characterised *S.* Typhimurium ST313 associated bloodstream infection in Malawi [4].

The remaining seven phages were isolated with several other host strains, including the *S.* Typhimurium ST19 representative strain 4/74, *S.* Typhimurium D23580 ΔΦ [8] (with the BREX system intact), and *S.* Enteritidis strains.

### 3.2. Host Range Analysis

The host range of all 32 phages was tested using bacterial strains representing *S.* Typhimurium ST19 and ST313, and *S.* Enteritidis Global Epidemic and African clades (Figure 1). No significant difference in host range was observed between phages isolated from the UK and those from Malawi (Figure 1). A small proportion of phages (9 phages, 28%) that originated from UK or Malawi were able to infect *S.* Enteritidis from the African clades, of which only two (BPS5 and BPS6) were isolated in Liverpool with *S.* Enteritidis as the host. BPS5 was the only phage capable of infecting all hosts tested, including mutants. The phages BPS3, BPS6 and BPS7 were able to infect all wild-type *Salmonella* strains tested. These results indicate that phages isolated in the UK had a broader host range than phages isolated from Malawi.

Of the 32 phages isolated, seven were unable to infect *S.* Enteritidis. The Global Epidemic *S.* Enteritidis strains P125109 and A1636 were susceptible to infection by the same phages, with the exception of phage ER8 that only infected A1636 (Figure 1). This result reflects the fact that these *S.* Enteritidis strains were very similar at the genomic level, differing by ~40 SNPs. *S.* Enteritidis from the African clades were resistant to infection of most (>70%) of the 32 phages. *S.* Enteritidis strain 4030/15, a representative from the West African clade, was susceptible to infection of the same phages as D7795 and CP255, representatives of the Central/Eastern African clade, with the addition of phages BPS1, ER2, ER4, ER5 and ER10.

*S.* Enteritidis strains, especially from the African clades, are also resistant to being infected by other commonly used bacteriophages, such as the *Salmonella* phages 9NA, BTP1 and P22 (Figure 1). The P22 transducing phage is commonly used for genetic manipulation of *Salmonella.* However, P22 infects *S.* Enteritidis from the Global Epidemic clade at a lower level than *S*. Typhimurium, and did not cause clearing of infected *S.* Enteritidis cultures due to an unknown mechanism. Interestingly, *Salmonella* phage Det7 was capable of infecting both *S*. Typhimurium and *S.* Enteritidis hosts, sharing a similar host range profile to phages BPS3, BPS5, BPS6, and BPS7.

More phages infected *S.* Typhimurium ST19 (Figure 1) strains 14028s (32 phages) and LT2 (29 phages), than *S.* Typhimurium 4/74 (eight phages), possibly due to prophage-encoded “anti-phage” defence mechanisms. Host range was tested on a *S*. Typhimurium 4/74 mutant that lacked prophages (4/74 ΔΦ), and the number of infecting phages increased from eight to 13 (Figure 1).

All isolated phages, except for ER21, were able to infect *S.* Typhimurium D23580 (Figure 1). However, in comparison to the original host D23580 strains that were deleted for either prophages (D23580 ΔΦ) or prophages and BREX (D23580 ΔΦΔ*brex*), some plaque sizes were smaller on D23580 wild type (Figure 1). This suggested phages ER22, ER23, ER25, BPS3, BPS6 and BPS7 are susceptible to a resistance mechanism encoded by one of the D23580 prophages. BREX did not appear to impact these phages (Figure 1).

Phage BTP1 is one of the key prophages carried by *S.* Typhimurium ST313 that has very high spontaneous induction (~10^9^ pfu/mL) [8]. BTP1 is active and highly conserved in both the lineages 1 and 2 of ST313 that are responsible for about two-thirds of the iNTS cases in Sub-Saharan Africa [53]. Here, we used a *S.* Typhimurium D23580 mutant (D23580 ΔBTP1) to show that BTP1 conferred resistance against phage ER21, possibly due to the removal of the novel BstA defence system [10] or the GtrAC LPS-modification enzymes [11].

Cell-surface properties frequently mediate phage-resistance by blocking attachment of tailed-bacteriophages, either by modification of the lipopolysaccharide (LPS), outer membrane proteins or flagella [54]. The LPS oligo-polysaccharide residues (O-antigen) are common phage receptors [55]. To determine whether the O-antigen was required for phage infection, a *S.* Typhimurium D23580 mutant lacking all prophages, the BREX system, and *galE* (D23580 ΔΦ Δ*brex* Δ*galE*) was tested. The gene *galE* encodes the UDP-glucose 4-epimerase component of the galactose biosynthetic pathway, and is required for the synthesis of the LPS O-antigen of *Salmonella*. Phages ER25, BPS3, BPS6, BPS7, ER6 and ER18 were unable to infect the mutant D23580 ΔΦ Δ*brex* Δ*galE* (Figure 1), suggesting that LPS is the receptor for attachment of these bacteriophages.

### 3.3. Comparative Genomics and Phylogeny of Isolated Phages

We used whole genome sequencing to investigate the taxonomy and relatedness of all the Malawian and UK phages. Pairwise genome comparison of nucleotide similarity between the 32 isolated phages revealed three main clusters (Appendix A). The relatedness of the bacteriophages was also assessed by alignment of the gene encoding the terminase large sub-unit (*terL*) that was present in all phages (Figure 2).

Cluster 1 contained phages isolated in the UK that were able to infect all wild-type *Salmonella* strains tested. Cluster 2 included a mixture of bacteriophages isolated in the UK and Malawi that were only capable of infecting *S.* Typhimurium (except phage BPS1). Cluster 3 was divided into two sub-clusters: 3.a contained phages isolated from the UK, and 3.b from Malawi. Generally, the Cluster 3 phages infected *S.* Typhimurium 14028s, LT2 and D23580, and *S.* Enteritidis from the Global Epidemic clade, with a few of the phages infecting the representative West African clade strain 4030/15. Phage ER25 was the only one that did not belong to Cluster 1, 2 or 3. Phages belonging to Cluster 1 had a relatively large genome of ~158 kb, compared to the ~60 kb genomes of Cluster 2 phages, and ~43 kb genomes of Cluster 3 phages.

Comparative genomic analysis of Cluster 1 phages (Figure 3A) showed that BPS3, BPS5 and BPS6 were closely related to S117 (>98% nucleotide identity; GenBank MH370370.1), a phage originally isolated on *S.* Typhimurium LT2 [56]. Figure 3B shows that most phage-encoded genes are conserved throughout Cluster 1. BPS3 lacked one of the tail-spike genes present in BPS6, BPS5, BPS7, and reference phage S117, a gene that is upstream of the predicted virulence protein VriC, also found in other *Salmonella* phages [57]; however, it is not clear whether VriC is a functional effector protein. Other genes were missing from Cluster 1 phages such as BPS7, that encode hypothetical proteins preventing any inference of their function. The comparative genomic analysis identified 26 SNP differences between BPS5 and BPS6, all of which are in intergenic regions upstream hypothetical proteins and tRNA genes. None of these nucleotide differences explained why BPS5 is capable of infecting all hosts tested, including the mutant *S.* Typhimurium D23580 strain with a short LPS (D23580 ΔΦ Δ*brex* Δ*galE*), but BPS6 could not. Interestingly, BPS3 is the only phage in Cluster 1 with a different tail-spike protein (tail-spike 1).

Cluster 2 (Figure 4A) included phages isolated from Malawi and the UK that were closely related (>93% identity) to *S.* Typhimurium phages iEPS5 (GenBank KC677662.1) [58] and øχ (Chi, GenBank NC_025442.1) [59], which uses the flagellum as the receptor, and phage 37 [60] isolated from sewage in India. Based on the comparative genomic analysis (Figure 4B), seven structural genes including a tape measure protein, head vertex, and a portal protein were identified. Additionally, genes encoding putative endolysins proteins A and B and terminase large and small subunits (*terSL)* were identified.

Cluster 3 was the largest and most diverse cluster, with different host range and genome composition. Genome comparison (Figure 5A) indicated a clear division between sub-clusters 3.a and 3.b, with sub-cluster 3.b only containing phages isolated from Malawi. Phages ER21, ER22 and ER23 (sub-cluster 3.a) (Figure 5B) were closely related (>96% identity) to phage S134 (GenBank MH370381) [56], originally isolated using *S*. Enteritidis PT1 strain E2331, which might indicate their ability to infect *S.* Enteritidis from the Global Epidemic clade. Sub-cluster 3.b included very closely related phages, with differences only in two hypothetical proteins (Figure 5C). Several structural genes were identified in phages from sub-cluster 3.a, however no structural genes were found in sub-cluster 3.b isolates, highlighting the novelty of these phages. Genes related to replication, like DNA polymerases and topoisomerases, and recombination were identified in phages from sub-cluster 3.b. Additionally, the lysozyme encoded by gene *rrrD* was identified; however, most genes were identified as hypothetical proteins.

Finally, phage ER25 did not belong to any of the identified clusters. Comparative genomic analysis demonstrated that phage ER25 is very similar (>99% identity) to P22 (GenBank AF217253.1) [61], differing by only 49 SNPs (Appendix A). This finding, complemented by analysis with PhageAI, suggests that ER25 was the only temperate phage identified, whereas all other isolated phages were likely to have a virulent life cycle.

Four bacteriophages were selected as representatives of each Cluster for morphological analysis by transmission electron microscopy. All phages were from the order *Caudovirales,* with two main morphotypes representing different phage families. Phages from Clusters 2 and 3 had a long (>100 nm) non-contractile tail and belonged to the *Siphoviridae* family, whereas the phages from Cluster 1 had a contractile tail, belonging to the *Ackermannviridae* family, confirmed by genome comparison with phage Det7 that also groups in Cluster 1.

### 3.4. Perspective

This study represents the first isolation of virulent phages that target *Salmonella* variants associated with bloodstream infection in Sub-Saharan Africa. The phages target multidrug resistant African lineages of iNTS that are particularly difficult to control by many antimicrobials and are not susceptible to well-characterised phages. Our data also demonstrate that prophage repertoire is critical for phage resistance in *Salmonella*, suggesting that novel anti-phage systems are likely to be discovered in the genomes of clinically relevant *Salmonella.* We have identified phages with different host ranges, including generalist and specialist phages against *S.* Typhimurium and *S.* Enteritidis. This phage collection has the potential for wider applications, such as decontamination of food, swine slurry [62], and hospital or food preparation surfaces [63], which is particularly relevant in low and middle-income countries [64], and could have utility against multidrug resistant iNTS infections in clinical settings.

## Figures and Tables

**Figure 1 viruses-13-00478-f001:**
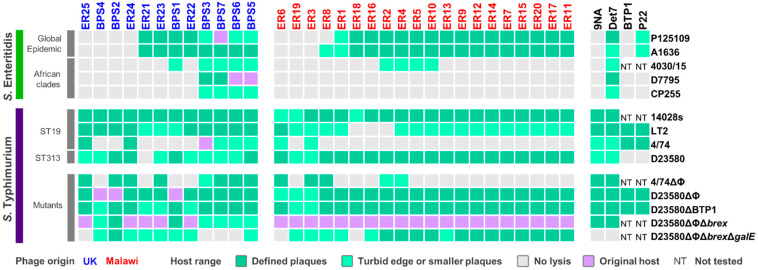
Host range of isolated phages. Host range analysis was determined by spot assay on *S.* Enteritidis or *S.* Typhimurium host strains, with phages isolated in the UK (blue) or Malawi (red). Reference phages are displayed in black. Host range of biological replicates was recorded as defined lysis (dark green), turbid edge or smaller plaques (light green), and no lysis (grey). Examples of the different lysis observed are shown in Appendix A. The original *Salmonella* host used for phage isolation is indicated in purple. NT: not tested.

**Figure 2 viruses-13-00478-f002:**
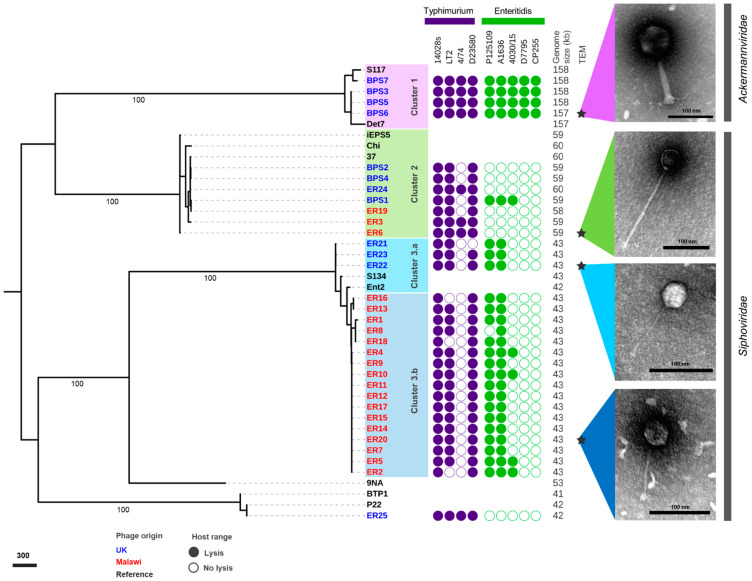
Phylogenetic and phenotypic characterisation of related isolated phages. Gene alignment of *terL* gene visualised in iTOL, with numbers indicating bootstraps on branches leading to clusters. The three clusters of related phages identified are presented in lilac (Cluster 1), green (Cluster 2), light blue (Cluster 3.a) and dark blue (Cluster 3.b). The host range for *S*. Typhimurium and *S*. Enteritidis is presented as a filled circle (lysis) or a clear circle (no lysis) from Figure 1, genome size (kb), and morphology observed by transmission electron microscopy (TEM) with scale bars of 100 nm. Inferred taxonomy is indicated next to TEM images. Phage names in blue were isolated from the UK and red from Malawi. Names in black indicate reference phages, and stars indicate representative phages used for TEM.

**Figure 3 viruses-13-00478-f003:**
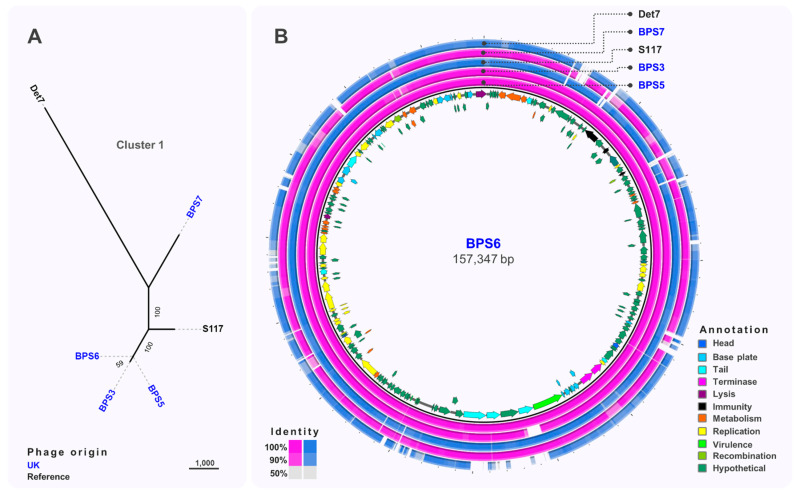
Comparative genomics of phages from Cluster 1. (**A**) Maximum-likelihood tree of phages in Cluster 1, unrooted and visualised in iTOL with numbers on branches indicating bootstraps. Scale bar represents number of single nucleotide polymorphisms. Phage names in blue were isolated from the UK and black font indicates reference phages. (**B**) Whole-genome comparison of phages in Cluster 1. The local similarity of each genome at the nucleotide level was calculated based on BLASTn high scoring pairs and plotted against a circular map of BPS6 represented as the inner circle. Blue circles represent reference genomes. Genome annotation is shown as arrows in the inner circle, colour-coded based on gene categories. Full annotation can be found in Appendix A.

**Figure 4 viruses-13-00478-f004:**
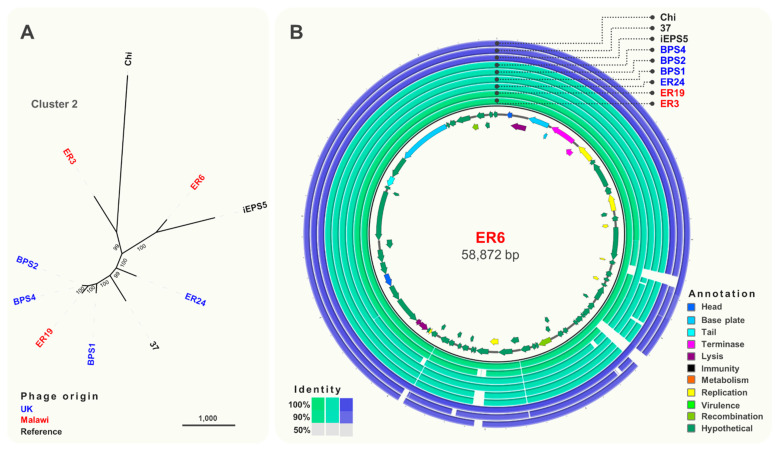
Comparative genomics of phages from Cluster 2. (**A**) Maximum-likelihood tree of phages in Cluster 2, unrooted and visualised in iTOL with numbers on branches indicating bootstraps. Scale bar represents number of single nucleotide polymorphisms. Phage names in blue were isolated from the UK and red from Malawi. Names in black font indicate reference phages. (**B**) Whole-genome comparison of phages in Cluster 2. The local similarity of each genome at the nucleotide level was calculated based on BLASTn high scoring pairs and plotted against a circular map of ER6 represented as the inner circle. Blue circles represent reference genomes. Genome annotation is shown as arrows in the inner circle, colour-coded based on gene categories. Full annotation can be found in Appendix A.

**Figure 5 viruses-13-00478-f005:**
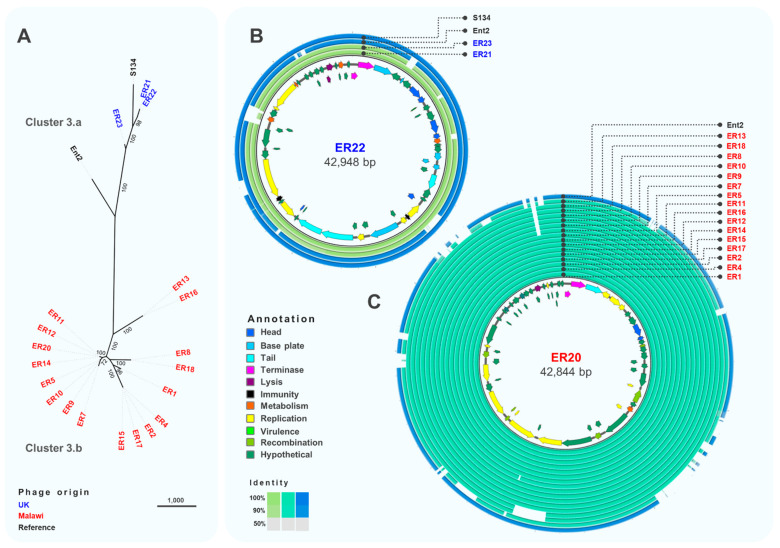
Comparative genomics of phages from Cluster 3. (**A**) Maximum-likelihood tree of phages in Cluster 3, unrooted and visualised in iTOL with numbers on branches indicating bootstraps. Scale bar represents number of SNPs. Phage names in blue were isolated from the UK and red from Malawi. Names in black font indicate reference phages. (**B**) Whole-genome comparison of phages in Cluster 3.a. The local similarity of each phage genome at the nucleotide level was calculated based on BLASTn high scoring pairs and plotted against a circular map of ER22, represented as the inner circle. Blue circles represent reference genomes. Genome annotation is shown as arrows in the inner circle, colour-coded based on gene categories. Full annotation can be found in Appendix A. (**C**) Whole-genome comparison of phages in Cluster 3.b. The local similarity of each genome was calculated based on BLASTn high scoring pairs and plotted against a circular map of ER20 represented as the inner circle. Blue circles represent reference genomes. Genome annotation is shown as arrows in the inner circle, colour-coded based on gene categories. Full annotation can be found in Appendix A.

**Table 1 viruses-13-00478-t001:** Published bacteriophages, bacteria strains, and plasmids used in this study.

Strain	Description ^a^	Reference
*Bacteriophages*		
P22	*Podoviridae*, temperate, wild type from strain LT2	[8,15]
P22 HT *105/1 int-201*	*Podoviridae*, virulent transducing phage	[16]
BTP1	*Podoviridae*, temperate, wild type from strain D23580	[8]
Det7	*Ackermannviridae*, virulent, wild type	[17]
9NA	*Siphoviridae*, virulent, wild type	[18]
*S. Enteritidis*		
P125109	Wild type	[19]
A1636	Wild type	[3]
4030_15	Wild type	[20]
D7795	Wild type	[3]
CP255	Wild type	[21]
*S. Typhimurium*		
14028s	Wild type	[22]
LT2	Wild type	[23]
*4/74 derivatives*		
4/74	Wild type	[24]
JH4180	4/74 ΔΦ (ΔGifsy-1 ΔGifsy-2 ΔST64B ΔSopEΦ ΔP4-like)	This study
*D23580 derivatives*		
D23580	Wild type	[4]
JH3877	D23580 ΔBTP1	[8]
JH3949	D23580 ΔΦ (ΔBTP1 ΔBTP5 ΔGifsy-1 ΔGifsy-2 ΔST64B)	[8]
JH4314	D23580ΔΦ Δ*brex::frt* (ΔSTMMW_44361- STMMW_44431)	This study
JH4655	D23580ΔΦ Δ*brex::frt* Δ*galE*	This study
*Escherichia coli*		
S17-1 λ*pir*	*pro thi hsdR recA* chromosome::RP4-2 Tc::Mu Km::Tn7/λ*pir*; Tp ^R^, Sm ^R^	[25]
*Plasmids*		
pEMG	Suicide plasmid; Km ^R^	[26]
pNAW1	pEMG::*attB*^Gifsy-2 4/74^; Km ^R^	This study
pNAW15	pEMG::*attB*^Gifsy-1^; Km ^R^	[8]
pNAW19	pEMG::*attB*^SopEΦ&P4-like^; Km ^R^	This study
pNAW34	pEMG::∆galE; Km ^R^	This study
pNAW42	pEMG::*attB*^ST64-B^; Km ^R^	[8]
pSW-2	I-*Sce*I expressing vector, *m*-toluate inducible; Gm ^R^	[26]
pKD4	*frt-aph-frt* (Km ^R^ cassette flanked with *frt*) template plasmid; Km ^R^	[27]
pSIM5-*tet*	λ Red recombination plasmid, temperature-inducible; Tc ^R^	[28]
pCP20-Gm	Flippase recombinase expression plasmid; Gm ^R^	[29]

^a^ Relevant antibiotic resistances are indicated by ^R^: Gm, gentamicin; Km, kanamycin; Sm, streptomycin; Tc, tetracycline; Tp, trimethoprim. Natural resistances of wild type strains are not indicated. The flippase recognition target sequence from pKD4 is indicated by “*frt*”.

**Table 2 viruses-13-00478-t002:** *Salmonella* phages isolated in this study and associated metadata.

Common Name	Phage Name	Place of Isolation	Original Host	Cluster ^a^	GenBank Accession Number
ER1	vB_SenS_ER1	Blantyre, Malawi	*S.* Typhimurium D23580ΔΦ*brex*	3.b	MW355461
ER2	vB_SenS_ER2	Blantyre, Malawi	*S.* Typhimurium D23580ΔΦ*brex*	3.b	MW355466
ER3	vB_SenS_ER3	Blantyre, Malawi	*S.* Typhimurium D23580ΔΦ*brex*	2	MW355467
ER4	vB_SenS_ER4	Blantyre, Malawi	*S.* Typhimurium D23580ΔΦ*brex*	3.b	MW355468
ER5	vB_SenS_ER5	Blantyre, Malawi	*S.* Typhimurium D23580ΔΦ*brex*	3.b	MW355469
ER6	vB_SenS_ER6	Blantyre, Malawi	*S.* Typhimurium D23580ΔΦ*brex*	2	MW355470
ER7	vB_SenS_ER7	Blantyre, Malawi	*S.* Typhimurium D23580ΔΦ*brex*	3.b	MW355471
ER8	vB_SenS_ER8	Blantyre, Malawi	*S.* Typhimurium D23580ΔΦ*brex*	3.b	MW355472
ER9	vB_SenS_ER9	Blantyre, Malawi	*S.* Typhimurium D23580ΔΦ*brex*	3.b	MW355473
ER10	vB_SenS_ER10	Blantyre, Malawi	*S.* Typhimurium D23580ΔΦ*brex*	3.b	MW355451
ER11	vB_SenS_ER11	Blantyre, Malawi	*S.* Typhimurium D23580ΔΦ*brex*	3.b	MW355452
ER12	vB_SenS_ER12	Blantyre, Malawi	*S.* Typhimurium D23580ΔΦ*brex*	3.b	MW355453
ER13	vB_SenS_ER13	Blantyre, Malawi	*S.* Typhimurium D23580ΔΦ*brex*	3.b	MW355454
ER14	vB_SenS_ER14	Blantyre, Malawi	*S.* Typhimurium D23580ΔΦ*brex*	3.b	MW355455
ER15	vB_SenS_ER15	Blantyre, Malawi	*S.* Typhimurium D23580ΔΦ*brex*	3.b	MW355456
ER16	vB_SenS_ER16	Blantyre, Malawi	*S.* Typhimurium D23580ΔΦ*brex*	3.b	MW355457
ER17	vB_SenS_ER17	Blantyre, Malawi	*S.* Typhimurium D23580ΔΦ*brex*	3.b	MW355458
ER18	vB_SenS_ER18	Blantyre, Malawi	*S.* Typhimurium D23580ΔΦ*brex*	3.b	MW355459
ER19	vB_SenS_ER19	Blantyre, Malawi	*S.* Typhimurium D23580ΔΦ*brex*	2	MW355460
ER20	vB_SenS_ER20	Blantyre, Malawi	*S.* Typhimurium D23580ΔΦ*brex*	3.b	MW355462
ER21	vB_SenS_ER21	Liverpool, UK	*S.* Typhimurium D23580ΔΦ*brex*	3.a	MW355463
ER22	vB_SenS_ER22	Liverpool, UK	*S.* Typhimurium D23580ΔΦ*brex*	3.a	MW355464
ER23	vB_SenS_ER23	Southport, UK	*S.* Typhimurium D23580ΔΦ*brex*	3.a	MW355465
ER24	vB_SenS_ER24	Durham, UK	*S.* Typhimurium D23580ΔΦ*brex*	2	MW355479
ER25	vB_SenP_ER25	Durham, UK	*S.* Typhimurium D23580ΔΦ*brex*	ND	MW355480
BPS1	vB_SenS_BPS1	Liverpool, UK	*S.* Typhimurium D23580ΔΦ	2	MW355474
BPS2	vB_SenS_BPS2	Liverpool, UK	*S.* Typhimurium D23580ΔΦ	2	MW355449
BPS3	vB_SenAc_BPS3	Liverpool, UK	*S.* Typhimurium 4/74	1	MW355475
BPS4	vB_SenS_BPS4	Liverpool, UK	*S.* Typhimurium D23580ΔΦ	2	MW355476
BPS5	vB_SenAc_BPS5	Liverpool, UK	*S.* Enteritidis D7795	1	MW355477
BPS6	vB_SenAc_BPS6	Liverpool, UK	*S.* Enteritidis D7795	1	MW355450
BPS7	vB_SenAc_BPS7	Liverpool, UK	*S*. Enteritidis P125109	1	MW355478

^a^ Clusters determined based on whole genome alignment. ND: not determined.

## Data Availability

The data presented in this study are publicly available in GenBank, with accession numbers listed in Table 2.

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
