# Peer review of "Isolation and Characterisation of Bacteriophages with Activity against Invasive Non-Typhoidal Salmonella Causing Bloodstream Infection in Malawi"

_viruses, 2021, doi:10.3390/v13030478_

Round 1
Reviewer 1 Report
The paper seem to be interesting. But I have a few answers:
1) Have you tested phage stability under different temperatures and NaCl concentrations?
2) Also, have you tested the application of phages cocktails to control Salmonella in food?
Reviewer 2 Report
The manuscript by Rodwell et al, "Isolation and characterization of bacteriophages with activity against invasive non-typhoidal Salmonella causing blood-stream infection in Malawi" describes a set of virulent bacteriophages capable of infecting a rather broad range of novel invasive non-typhoidal Salmonella (iNTS) stains associated with blood-stream infections prevalent in sub-Saharan Africa. Such phages might offer alternative treatments for multi-resistant strains and be useful e.g. for the decontamination of food.
Overall the paper is very well written, with a very (sometimes maybe even too?) detailed Materials and Methods, not that common these days! The new phages form 4 clusters (and a singleton), all of which (but one) contain one previously identified Salmonella phage, added to explore the diversity of these new isolates. The analysis also revealed the important role of Salmonella prophages in the resistance of their host to phage infection, a possible indication of new phage-resistance mechanisms to be discovered. Minor revision (corrections?) may be needed before publication, essentially in the figures. They are listed below, going along the text, regardless of their relative importance.
Lines 50-51: "Prophages found in iNTS have been described in S. Typhimurium [8,9], and partially described in S . Enteritidis [3]." This sentence lacks precision. How many strains, how many prophages? And related to the next sentence, has their role in fitness and preventing infection been assessed or was this shown with other prophages and strains?
Line 195 and 202: briefly outline what PhageAI and Roary are and why the latter was chosen for this small set of genomes.
Line 223: Malawi,
Lines 304-305: not sure I fully understand this: what do you mean by high diversity? by gene presence-absence? Are these genes nucleotide sequences related? How were genes with no similarity accounted for?
Legend Figure 2: As for above, "A gene presence/absence tree was constructed based on binary genes using Roary and visualized in iTOL" is hardly understandable. Compared to the rest of the Materials and Methods the part related to this is really cryptic! The choice of the colors in the figure is weird. Not sure light green is really green, or blue? Why don't reference phages have a host range? Is everyone familiar with the family "Ackermanviridae"... this also for line 416: contractile tail, particular head size? Other features?
Line 328: I'm surprised by the fact that TerS was used along with TerL. Isn't TerS much less conserved than TerL and is there a conservation of TerS-TerL pairs?
Line 337: Why isn't S117 shown in Figure 3?
Figure 4: not clear what pink and blue circles refer to? Here again the choice of colors isn't optimal The difference between immunity and lysis or base-plate and tail for instance is hard to distinguish on the circular map. And if there is an "immunity" how does this reconcile with the virulence of the phage (line 410)? These comments hold for Figure 5 and 6 as well.
Figure 6: see above and possibly add 3a and 3b in panel A. Isn't there an inversion between these a and b and those in Figure 3?
Lines 406-410: with no indication on how PhageAI estimates (I suppose) probabilities for a phage to be temperate or virulent this is not a robust conclusion!
Line 425: to be discovered
Figure S1 and text: "indistinct plaques" isn't the best decsription of what is shown in the figure. The size is distinct, the clear center as well... to me indistinct would have meant fuzzy and of varying sizes, which isn't the case here.
Last point: really strange this "reisolation" of P22. Are you sure nobody was using the phage near the site of sampling? This reminds me of a phage course at CSHL where T1 and T4 were isolated from sewage water out of the University of Wisconsin in Madison
Reviewer 3 Report
The manuscript entitled " Isolation and characterisation of bacteriophages with activity against invasive non-typhoidal Salmonella causing blood-stream infection in Malawi" hits important pains related to salmonella-mediated bloodstream infection in the less wealthy part of the world. The manuscript is written in excellent scientific style.
Regarding the methodology section, please clarify why the samples were incubated for 3 days? I think this results in the formation of numerous phage-insensitive mutants.
The host range determination should base on EOP methodology due to the generation of false-positive results [https://dx.doi.org/10.1371%2Fjournal.pone.0118557]. This is not a mistake, but You could consider this method for Your future work.
Some methods section are well known for phage scientists and should be shortened.
Discussion of the potential method of salmonella-infection prevention should be more developed. You mention food decontamination. You might add use for slurry decontamination [https://doi.org/10.1016/j.apsoil.2017.06.020] and surface decontamination [https://dx.doi.org/10.4161%2Fbact.25697], Also, hospital surfaces.
